# Adiponectin Deregulation in Systemic Autoimmune Rheumatic Diseases

**DOI:** 10.3390/ijms22084095

**Published:** 2021-04-15

**Authors:** Neža Brezovec, Katja Perdan-Pirkmajer, Saša Čučnik, Snežna Sodin-Šemrl, John Varga, Katja Lakota

**Affiliations:** 1Department of Rheumatology, University Medical Centre Ljubljana, 1000 Ljubljana, Slovenia; katja.perdanpirkmajer@kclj.si (K.P.-P.); sasa.cucnik@kclj.si (S.Č.); snezna.sodin@kclj.si (S.S.-Š.); katja.lakota@kclj.si (K.L.); 2Faculty of Pharmacy, University of Ljubljana, 1000 Ljubljana, Slovenia; 3Faculty of Medicine, University of Ljubljana, 1000 Ljubljana, Slovenia; 4FAMNIT, University of Primorska, 6000 Koper, Slovenia; 5Department of Internal Medicine, University of Michigan, Ann Arbor, MI 48109, USA; vargaj@med.umich.edu

**Keywords:** adiponectin, systemic autoimmune rheumatic diseases, therapy, gene regulation, interleukin 6, tumor necrosis factor α, PPAR-γ

## Abstract

Deregulation of adiponectin is found in systemic autoimmune rheumatic diseases (SARDs). Its expression is downregulated by various inflammatory mediators, but paradoxically, elevated serum levels are present in SARDs with high inflammatory components, such as rheumatoid arthritis and systemic lupus erythematosus. Circulating adiponectin is positively associated with radiographic progression in rheumatoid arthritis as well as with cardiovascular risks and lupus nephritis in systemic lupus erythematosus. However, in SARDs with less prominent inflammation, such as systemic sclerosis, adiponectin levels are low and correlate negatively with disease activity. Regulators of adiponectin gene expression (PPAR-γ, Id3, ATF3, and SIRT1) and inflammatory cytokines (interleukin 6 and tumor necrosis factor α) are differentially expressed in SARDs and could therefore influence total adiponectin levels. In addition, anti-inflammatory therapy could also have an impact, as tocilizumab treatment is associated with increased serum adiponectin. However, anti-tumor necrosis factor α treatment does not seem to affect its levels. Our review provides an overview of studies on adiponectin levels in the bloodstream and other biological samples from SARD patients and presents some possible explanations why adiponectin is deregulated in the context of therapy and gene regulation.

## 1. The Adiponectin Paradox

Adiponectin expression is downregulated by inflammatory mediators such as interleukin 6 (IL-6) and tumor necrosis factor α (TNF-α). Accordingly, decreased serum levels have been described in diseases with low-grade chronic inflammation such as type 2 diabetes, obesity, and atherosclerosis, and adiponectin deficiency has been linked to the pathogenesis of these diseases [1,2]. Paradoxically, high adiponectin levels are associated with various inflammatory diseases such as rheumatoid arthritis (RA), despite persistent inflammation, and have been associated with general and cardiovascular mortality [1]. Thus, it appears that the optimal window of adiponectin concentration is tight and any deregulation leads to a pathology. However, the explanation why adiponectin expression and regulation is different in some inflammatory diseases is not known. Resolving this paradox could provide an important insight into the pathogenesis of these diseases and clarify how adiponectin may be involved, potentially making it a treatment target.

A majority of up-to-date reviews addresses the association between circulating adiponectin levels and specific immune-mediated, rheumatic, or connective tissue diseases [3,4,5,6,7], or focuses on the role of adiponectin in the pathogenesis of these diseases [1,8,9,10].

In this review, we will examine the deregulation of adiponectin in the setting of several systemic autoimmune rheumatic diseases (SARDs) to try to provide some possible explanations for the paradoxically elevated levels in highly inflammatory SARDs. We will present adiponectin levels in circulation and in other biological samples in addition to associations with clinical manifestations of the diseases. Our review is the first in the field to highlight the effects of various drugs used in SARDs on circulating adiponectin levels and to suggest how inflammatory cytokines, such as interleukin-6 and tumor necrosis factor α, and gene regulation might contribute to deregulated adiponectin expression in SARDs.

### 1.1. Adiponectin Expression, Structure, Isoforms and Signaling

Adiponectin is the most abundant circulating adipokine in a human. It is present in serum at 0.01% to 0.05% of total proteins and plasma concentrations of 4 to 37 µg/mL [11]. They are in a range of about a thousand times higher than that of other hormones (e.g., insulin, leptin) and closer to the concentrations of some carrier proteins (e.g., retinol-binding proteins) [12]. Most adiponectin is produced by healthy adipose tissue, which is why it is classified as adipokine. Other tissues (Figure 1) and cells, such as skeletal myocytes, cardiomyocytes, osteoblasts, liver parenchyma cells, pituitary and endothelial cells may also be a source [13,14].

The structure of adiponectin consists of four parts, the N-terminal signal sequence, the variable region, the collagenous domain and the C1q-like globular domain [11]. After synthesis, adiponectin undergoes further post-translational modifications to form low molecular weight (LMW) trimers, medium molecular weight (MMW) hexamers or multimeric high molecular weight (HMW) isoforms (Figure 2a). They are secreted and bind differently to adiponectin receptors [15]. AdipoR1 and AdipoR2 are the best characterized receptors, while binding to T-cadherin and calreticulin is less studied. After binding to AdipoR1/R2, adiponectin stimulates two main signaling pathways, through PPAR-α and AMPK. Their activation and the activation of other downstream molecules leads to metabolic effects (e.g., fatty acid oxidation, glucose uptake), vasodilatation, as well as reduced apoptosis, inflammation, and fibrosis (Figure 2b). In AdipoR, pockets with ceramidase activity capable of metabolizing sphingosine to sphingosine-1-phosphate and thereby reducing cellular ceramide content were discovered [11].

### 1.2. What Affects Adiponectin Levels?

Adiponectin levels are influenced by several physiological, pathological and external factors. They are dependent on ethnicity and correlate negatively with body mass index, but positively with age. High levels of adiponectin are present in centenarians, where they may act as a compensatory response in maintaining metabolic and redox homeostasis. Interestingly, minimal levels are observed early in the morning, suggesting that the circadian rhythm may also be involved in regulation. In addition, sexual dimorphism is observed, with higher total and higher HMW adiponectin concentrations in women. This could be due to the inhibitory effect of testosterone on HMW adiponectin production [11], and due to the fact that females (after puberty) generally have a higher percentage of body fat than males [16]. Adiponectin expression is affected by endoplasmic reticulum stress, oxidative stress [17] and β-adrenergic activation [15]. It has been shown that some therapeutic agents, e.g., antidiabetic agents thiazolidinediones (TZD), also affect adiponectin levels [11].

Adiponectin is expressed in healthy adipocytes, whereas adipose tissue in obesity is hypoxic and inflamed, and as such adipocytes produce less adiponectin [18]. Reduced adiponectin levels are also observed in obesity-related diseases such as type 2 diabetes, cardiovascular disease (CVD) [13] and obesity-associated cancers [19]. Another illustration of this phenomenon are diseases in which adipose tissue is reduced, such as anorexia nervosa, and these usually have elevated serum adiponectin levels [20]. It should be noted that the local loss of adipocytes can also have a major impact on disease pathogenesis. In the case of systemic sclerosis (SSc), dedifferentiation of adipocytes in the intradermal fat depot leads to a local decrease in adiponectin synthesis. The inhibitory effect of adiponectin on fibroblasts is lost, allowing the excessive production of the extracellular matrix, which leads to fibrosis [21,22].

## 2. Adiponectin in Systemic Autoimmune Rheumatic Diseases

Systemic autoimmune rheumatic diseases (SARDs) are heterogeneous disorders with prominent autoimmune dysregulation leading to multiorgan involvement and varying degrees of inflammation. Their etiology is unknown, but a genetic predisposition has been found [22,23]. SARDs include RA (the most frequent disease), connective tissue diseases such as systemic lupus erythematosus (SLE), Sjögren’s syndrome (SS), systemic sclerosis (SSc), and antiphospholipid syndrome (APS). Since there is no cure for these diseases, long-term pharmacotherapy is required and significant morbidity and mortality are observed. There are many studies focusing on adiponectin serum levels in various SARDs. The general circulating values calculated in meta-analyses of some of these are presented in Figure 3.

### 2.1. Rheumatoid Arthritis

RA is characterized by chronic inflammation of the synovium, leading to joint deformation. Small and large joints are affected, characterized by synovial hyperplasia, bone edema progressing into a joint space and bone erosion. Systemic inflammation and extra-articular manifestations may also occur affecting the skin, eyes, heart, lung, kidneys, and nervous and gastrointestinal systems. Additionally RA patients have a high risk of CVD [24]. Obesity increases the probability of developing RA and is associated with poorer disease outcomes and impaired responses to treatment [25].

#### 2.1.1. Serum

A meta-analysis from 2017 (12 studies, RA *n =* 784, healthy controls (HCs) *n =* 655) revealed that circulating adiponectin levels are significantly higher in RA patients than in HCs (SMD = 1.5) [7]. Several additional studies published later (RA *n =* 168, HC *n =* 90) showed a similar trend [26,27,28] and some found no difference (RA *n =* 131, HC *n =* 131) [29,30], while one study reported lower levels (RA *n =* 95, HC *n =* 95) [31]. However, adiponectin levels were not changed between RA (*n =* 820) and osteoarthritis or undifferentiated arthritis controls (*n =* 298), [30,32,33,34,35]. Also, no changes were seen when comparing only early RA patients (*n =* 97) and HC (*n =* 96) [29,36].

Circulating adiponectin levels in RA are associated with several clinically relevant characteristics as shown in Table 1. RA disease activity, most commonly measured as Disease Activity Score of 28 joints (DAS28), showed positive correlations with serum adiponectin levels in the majority of the studies. Higher adiponectin levels are associated with radiographic severity and progression and are independently related to poorer bone outcomes and reduced muscle mass. However, the status of adiponectin in carotid atherosclerosis is not clear. Lower serum levels of adiponectin were found in RA smokers compared to never smokers [37]. Two studies reported lower adiponectin in obese RA patients [38,39], while one found similar levels compared to non-obese patients [40]. In a cohort of subjects who had suffered from obesity (followed for up to 29 years), high serum levels of adiponectin at baseline were associated with an increased risk for RA [41].

#### 2.1.2. Synovial Fluid

Adiponectin levels in the synovial fluid of RA patients (*n =* 39) were lower than in serum and correlated positively with disease activity [46]. Specifically, compared to serum levels the HMW adiponectin levels in synovial fluid were lower but the LMW levels were higher. No significant difference was found for MMW levels (*n =* 7) [58].

#### 2.1.3. Cells/Tissues

Synovium with articular adipose tissue, in particular synoviocytes and articular adipocytes from RA and osteoarthritis patients, express adiponectin strongly both at the transcriptional and the protein level [59]. In RA tissue explants, the synovial membrane produces twice as much adiponectin as the articular adipose tissue regardless of activation status [60]. Also, subcutaneous abdominal adipose tissue of RA patients secretes more adiponectin than the corresponding osteoarthritis tissue [33,34].

### 2.2. Systemic Lupus Erythematosus

SLE is characterized by the formation of immune complexes that are deposited in the body and cause inflammation and complement activation. The heterogeneous presentation of the disease includes neuropsychiatric, gastrointestinal, hematologic, renal, pulmonary and cardiovascular manifestations and cardiovascular disease (CVD) occurs in more than 50% of SLE patients. With atherosclerosis being an important long-term complication, patients tend to develop carotid intima media thickening (IMT) and carotid plaques [61]. Lupus nephritis (LN), a form of glomerulonephritis, is one of the most serious organ involvements in SLE, and despite the advances in understanding its pathology and in improving treatment options, it remains a significant cause of death in SLE patients [62]. Disease activity is measured by the Systemic Lupus Erythematosus Disease Activity Index (SLEDAI) [63].

#### 2.2.1. Serum

Circulating adiponectin levels are significantly higher in SLE than in HC (SMD = 0.547), as shown by a meta-analysis from 2020 (12 studies, SLE *n =* 1024, HC *n =* 720) [4]. Disease activity did not seem to correlate with serum adiponectin as evidenced by a meta-analysis in 2017 (SLE *n =* 782, HC *n =* 550) [4], but higher concentrations were associated with the presence of LN and correlated with the severity of proteinuria [64,65,66,67].

Studies conducted in SLE generally show a positive association of adiponectin with atherosclerotic development. Higher concentrations were found in patients with carotid plaque (plaque *n =* 118, no plaque *n =* 186) [68,69,70]; however, two studies from the same research group could not demonstrate any association [71,72]. A positive correlation between adiponectin levels and the presence of carotid plaque was independent of age, disease duration or SLE treatment [68]. Therefore, adiponectin, in combination with other markers, may be a potential biomarker for plaque prediction. In SLE patients with carotid IMT ≤ 0.8, adiponectin levels corresponded with IMT [73], while a negative correlation between adiponectin and vascular stiffness parameters was found [74].

#### 2.2.2. Urine

Several studies suggest that adiponectin is a potential urine biomarker to discriminate LN SLE patients. Adiponectin levels and the adiponectin-to-creatinine ratio are significantly higher in the urine of SLE patients with renal involvement (renal involvement *n =* 25, no renal involvement *n =* 25) [75] and in active LN SLE (*n =* 33) compared to active non-LN SLE (*n =* 16), or patients with only LN history (*n =* 30) [76]. Similarly, adiponectin urine concentrations are significantly elevated in active LN (*n =* 125) compared to inactive LN (*n =* 31), SLE without LN (*n =* 36), or HC (*n =* 55) [77]. The adiponectin-to-creatinine ratio in patients with LN (*n =* 27) was also increased compared to normal controls (*n =* 8) [66]. There were differences in urine adiponectin levels based on LN activity status, as observed in a renal biopsy; levels were lower in patients with low-to-moderate LN activity than in patients with high LN activity [78,79]. Adiponectin is also included as a biomarker for the calculation of the Pediatric Renal Activity Index for Lupus (p-RAIL), which reflects histological LN activity [78,80].

#### 2.2.3. Cells

Adiponectin expression in peripheral blood mononuclear cells (PBMCs) was significantly higher in SLE patients (*n =* 46) compared to HC (*n =* 51). However, there were no differences in expression between SLE patients with and without LN and no associations with major clinical and laboratory parameters [81].

### 2.3. Ankylosing Spondylitis

Ankylosing spondylitis (AS) is a disease of the sacroiliac joint and the spine and the adjacent soft tissues such as tendons and ligaments. It is associated with the presence of the major histocompatibility complex class I allele HLA-B27 and the interleukin 23/17 axis [82].

#### Serum

Adiponectin levels do not differ significantly in AS patients compared to HC, as concluded by meta-analysis in 2017 (6 studies, AS *n =* 273, HC *n = 202*) [6] and one additional study (AS *n =* 20, HC *n =* 11) [83]. It needs to be mentioned that the lack of significance might be due to the sample size because the meta-analysis showed SMD = 0.460. Adiponectin levels did not change in longitudinal monitoring over the two-year period [83], and no correlation with disease activity or functional indices was found [84]. Clinically, baseline serum levels were lower in patients who showed radiographic progression of the spine after two years. There was also a significant inverse correlation between radiographic progression of the spine and HMW adiponectin isoform levels [85].

### 2.4. Systemic Sclerosis

SSc is characterized by microvascular damage, changes in the immune system and extensive fibrosis of the skin and organs such as lungs, kidneys, gastrointestinal tract and heart. Skin fibrosis, usually quantified by the modified Rodnan skin score (mRSS), is a distinguishing feature of SSc [86]. Pulmonary complications are another hallmark, as about 40% of patients develop interstitial lung disease (ILD); about 15% have pulmonary arterial hypertension; and the majority of patients also have pathological changes to the gastrointestinal tract [86]. The early stage of the disease is usually characterized by microvascular pathology that can manifest itself as skin ulcers, and the late stage is characterized by abnormal fibroblast activation leading to tissue fibrosis [87]. Based on the extent of skin fibrosis and involvement of internal organs, SSc patients can be divided into two subgroups: limited cutaneous SSc (lcSSc) and diffuse cutaneous SSc (dcSSc), the latter being associated with the more severe disease course and poorer outcomes [86].

#### 2.4.1. Serum

Circulating adiponectin levels are lower in SSc patients compared to HC (SMD = −0.638), as confirmed by a meta-analysis published in 2017 (11 studies, SSc *n =* 511, HC *n =* 341) [88] and a later published study (SSc *n =* 100, HC *n =* 20) [89]. The same meta-analysis showed, although not significantly, associations of low adiponectin levels in Caucasian and Asian ethnic groups [88]. Additionally, two other studies (SSc *n =* 246), found that patients with dcSSc had significantly lower concentrations than patients with lcSSc or HC [88,90,91,92]; thus, the downregulation in certain studies depended on the percentage of dcSSc and lcSSc patients included in each study. In dcSSc serum, adiponectin was the seventeenth most downregulated protein, out of 228 determined proteins [93].

Negative correlations between adiponectin levels and disease activity and progression, measured as Valentini disease activity index [94] and mRSS, were reported [87,95,96], with only one study reporting a positive correlation between adiponectin and mRRS in dcSSc patients [97]. One study found that patients with reduced concentrations showed a higher prevalence of pitting scars and pulmonary fibrosis [96].

The results on differences in adiponectin levels between early versus late SSc are not consistent. The majority (early SSc *n =* 49, late SSc *n =* 99) agree that the levels are lower in early SSc [95,97]. However, two studies reported either an opposite trend (early SSc *n =* 20, late SSc *n =* 16) [98] or no differences at all (early SSc *n =* 13, late SSc *n =* 16) [94]. The results are difficult to compare due to varying definitions of early SSc and late SSc from <18 months to 5 years for early SSc and >2 to 5 years of disease duration for late SSc. In addition, some studies included all SSc patients, while others focused only on dcSSc patients.

#### 2.4.2. Skin

SSc skin biopsies were examined for the expression of adiponectin and adiponectin-regulated genes. Mean relative transcript for adiponectin in skin tissue differed in the SSc subsets because the values in dcSSc patients (*n =* 5) were reduced compared to lcSSc patients (*n =* 7) and HC (*n =* 7) [96]. An inverse correlation of mRNA adiponectin levels with mRSS was found in lesional skin biopsies of early dcSSc (*n =* 15) and lcSSc patients (*n =* 6) [95]. Dermal levels of cellular phosphorylated AMPK, a molecule downstream of adiponectin signaling, were significantly reduced in SSc skin (*n =* 19) compared to HC (*n =* 4). In addition, lower or absent phosphorylated AMPK in myofibroblasts, were observed in the lesional skin of patients. Changes in gene expression were also found in SSc skin (*n =* 70) compared to HC (*n =* 22), with reduced scores of adiponectin signaling pathways [22].

Skin fibrosis is usually accompanied by a consistent decrease in dermal white adipose tissue (dWAT), an important source of adiponectin. The SSc patient skin biopsies showed a reduction in adipocyte number and size, as well as dWAT replacement by a fibrous matrix. This caused a local decrease in adiponectin synthesis, which resulted in the loss of its inhibitory signals on fibroblasts, allowing their activation into myofibroblasts. Adipocytes may be an important source of myofibroblasts as they are capable of undergoing adipocyte mesenchymal transition (AMT) (Figure 4). In this process, adipocytes lose expression of their distinct markers and begin to express myofibroblast markers, such as α-SMA, suggesting that they actively contribute to the extensive accumulation of extracellular matrix components that lead to fibrosis [21,99].

#### 2.4.3. Other

Overall, adiponectin expression in histologically stained lung tissue samples from patients with SSc and patients with idiopathic pulmonary fibrosis was significantly reduced compared to controls. However, no reduction of adiponectin was found in the study of early stage fibrotic bronchoalveolar lavage and lung protein lysates. In the same study, adiponectin was examined in gastroscopic biopsies, and they showed lower adiponectin levels in SSc gastritis compared to gastritis not associated with SSc [100].

### 2.5. Sjögren Syndrome

In Sjögren syndrome (SS), inflammation of exocrine glandular tissue, usually the lachrymal and salivary glands is present, leading to xerostomia, keratoconjunctivitis sicca and enlargement of the parotid gland. Salivary glandular epithelial cells (SGEC) are important in the pathogenesis of SS [101]. SS often overlaps with other SARDs and may also affect other organ systems and cause polyarthritis, cutaneous vasculitis, peripheral neuropathy, lung disorders, nephritis, optic neuritis, multiple sclerosis-like diseases. In addition, SS patients have an increased risk of lymphoma. Histopathologically, focal lymphocytic infiltrates, which are mainly located around the glandular ducts, are seen [102].

#### 2.5.1. Serum

Studies of serum concentrations of adiponectin in SS patients are scarce and they report either similar or higher circulating levels. Concentrations of adiponectin in primary SS (*n =* 29), SS associated with rheumatoid arthritis (*n =* 30), patients with non-autoimmune sicca syndrome (*n =* 17), and HC (*n =* 15) were at similar levels [103]. However, in another study, concentrations were higher in primary SS patients (*n =* 71) than in HC (*n =* 71) [102].

#### 2.5.2. Salivary Gland and Saliva

Adiponectin expression in minor salivary glands is among the highest tissue expression levels in the organism as shown in Figure 1. Adiponectin is additionally upregulated in the SGEC of SS patients, exerting a protective function against apoptosis [104]. In saliva, adiponectin levels normalized to total protein are higher in patients with SS (*n =* 17) than in HC (*n =* 13) or non-SS sicca patients (*n =* 19) and correlate with the xerostomia inventory, a scale for evaluating dry mouth [105].

### 2.6. Psoriatic Arthritis

Psoriatic arthritis (PsA) is an inflammatory arthropathy that affects some patients diagnosed with psoriasis, an immune-mediated disease of the skin and nails. It is the most common form of peripheral spondyloarthritis and it is associated with increased mortality from CVD. Obesity is an important comorbidity in PsA patients, and adiponectin may play a role in PsA, along with other pro- and anti-inflammatory mediators [106].

#### Serum

Results regarding circulating adiponectin levels in PsA are not consistent. PsA patients (*n =* 28) had significantly higher serum levels than HC (*n =* 39) in the study of Dikbas et al. [107]. Xue et al. proved the opposite, with PsA patients (*n =* 41) having lower adiponectin levels than HC (*n =* 24) and psoriasis patients (*n =* 20), and a negative correlation with osteoclast precursor numbers [108]. In another study PsA patients (*n =* 203) had higher adiponectin serum levels than patients with psoriasis without arthritis (*n =* 155) [109]. Finally, there were two studies in which no difference was found between PsA (*n =* 109 and *n =* 77, respectively) compared to HC (*n =* 32) [110] and osteoarthritis groups (*n =* 76) [111].

### 2.7. Antiphospholipid Syndrome

Antiphospholipid syndrome (APS) is a disorder, characterized by thrombosis, miscarriages and other pregnancy-related complications, in combination with persistent presence of antiphospholipid antibodies, among which are anti-β2GPI antibodies. Hematological, cutaneous, non-thrombotic cardiac and pulmonary, neurological, and renal manifestations may also be present [112].

#### Serum

In a single study, circulating adiponectin levels in primary APS patients (*n =* 56) were slightly but not significantly higher than in HC (*n =* 72). There was a positive correlation with anti-β2GPI IgG concentration [113].

In summary, circulating adiponectin levels are higher in RA and SLE, lower in SSc, while in AS patients the increase is not statistically significant. Due to a lack of studies and inconsistent results, the levels in SS, PsA and APS are still not well defined. Serum adiponectin in RA is positively associated with clinical manifestations, such as disease activity, radiographic severity and progression, poorer bone outcomes and reduced muscle mass. Negative associations are seen in SSc with disease activity and in AS with spinal radiographic disease progression. The role of adiponectin in CVD is not clear, there are studies that link adiponectin to increased mortality in CVD [2], and studies that report of its vasoprotective role [13]. A positive correlation of adiponectin with atherosclerotic plaques in SLE was found, which places SLE along studies reporting unfavorable adiponectin CVD effects. Unexpectedly, the association with CVD is not clear in RA. In general, adiponectin levels in the disease-affected tissues and cells reflect the levels in the bloodstream. Adiponectin is strongly expressed in RA synoviocytes and articular adipocytes, also levels in synovial fluid are associated with disease activity. The concentrations in urine strongly correlate with lupus nephritis in SLE, while in SS it is strongly expressed in the salivary gland, and normalized salivary levels correlate with dry mouth. In SSc, adiponectin expression is reduced not only in serum, but also in organs such as the skin, lung and gastroscopic sample tissue with less adiponectin present in the dcSSc subset with poorer clinical outcomes.

## 3. Could SARDs Treatment Change Serum Adiponectin?

Since SARDs are chronic diseases, patients need lifelong treatment which might affect adiponectin serum levels. We have investigated publications since 2010 collecting the effects of therapy on circulating adiponectin levels in SARDs patients. Most patients have received multidrug therapy, making the effects of a particular drug difficult to judge.

### 3.1. Anti-TNF

Anti-tumor necrosis factor (TNF) drugs were the first targeted biologicals approved for RA. Today, their use has spread to other chronic inflammatory diseases such as Crohn’s disease, ulcerative colitis, psoriasis, psoriatic arthritis, ankylosing spondylitis and juvenile RA [114].

As presented in Table 2, the majority of studies on RA, PsA and AS exhibit no change in adiponectin levels when anti-TNF therapy was used. However, three RA studies reported an increase in adiponectin. In the first study, the increase was presented only as the ratio of serum adiponectin to the body mass index [115]. The second study reported an increase at 3–12 months, but not during the 24-month period following treatment [116]. In the last report, an increase was only observed in patients who responded to treatment, but the measured adiponectin levels were 1000 times lower than in other reports [117].

Interestingly, the 3D-protein structure of the adiponectin C1q domain proved to be homologous to the TNF-α structure (Figure 5) in their identical folding topologies, key residue conservations and similarity of trimer interfaces. However, no homology was found in the primary amino acid sequence [128,129]. While anti-TNF therapy might increase adiponectin levels by inhibiting the inflammatory TNF-α signaling, the similar structure raises the question whether anti-TNF agents could bind to adiponectin and cause inactivation or reduction in circulation levels. Because serum levels after anti-TNF therapy remained unchanged, anti-TNF therapy did not seem to interfere with adiponectin detection. A study on AS patients treated with infliximab showed that adiponectin levels remained unchanged 120 min after therapy [127], suggesting that the clearance of adiponectin due to anti-TNF did not occur either. Also in vitro experiments confirmed that anti-TNF (adalimumab, etanercept) do not bind to adiponectin. However, they alter its signaling as shown in RA synovial fibroblasts (RASF), where adiponectin induction of IL-6 and matrix metalloproteinase-1, was reduced with application of anti-TNF. The preincubation of anti-TNF with adiponectin did not reduce this effect [59]. As shown later, RASF stimulated with adiponectin also did not increase TNF-α [130]. This suggests that observed anti-TNF effects inhibiting adiponectin proinflammatory signaling may be due to the binding of anti-TNF to transmembrane TNF-α, eliciting reverse transmembrane signaling.

### 3.2. Glucocorticoids

Glucocorticoids (GC) in SARDs are used as very effective tools for reducing disease activity and achieving clinical remission in the short term and reducing structural progression, disability and systemic manifestations in the medium term. They are widely used but have major drawbacks because of side effects [131,132]. Although they have been extensively studied in vitro, in animal models and in clinical trials, there is no consensus on the effects of GC on adiponectin expression [133]. In particular, GC is usually not used as a monotherapy, so its direct effect on adiponectin levels is difficult to discern. However, according to the studies, adiponectin serum levels generally tended to increase with GC treatment, but no changes were observed in the use of long-term, high-dose combination therapy. Elevated adiponectin levels after combined treatment with GC were associated with improved insulin resistance and endothelial function with a better lipid profile, and adiponectin may therefore play a role in cardiovascular protection (Table 3) [134,135].

There are two possible explanations for the association we see between GC and adiponectin levels, the first being the indirect GC affect through the decrease in inflammatory mediators. The second could be influenced by a GC-receptor binding site on the adiponectin gene promoter found in several cistromic studies in mice and humans [136]; however, this has not yet been well characterized or confirmed by other experiments.

### 3.3. Disease-Modifying Antirheumatic Drugs

Non-biological disease-modifying antirheumatic drugs (DMARDs) are a group of drugs that includes methotrexate (MTX), sulfasalazine (SSA), leflunomide, hydroxychloroquine (HCQ), azathioprine and some others. They are widely used in treatment of rheumatic diseases and are often prescribed in combination [138,139].

The increase in adiponectin after DMARD treatment was seen in most of the studies with RA patients. One report included patients (*n =* 27) treated with DMARDs in combination with prednisolone (the dose was reduced to 5 mg/d within one month) for 3 months [32]. Another study (*n =* 14) showed an increase in 24-month treatment with MTX to up to 20 mg/week with the addition of sulfasalazine 2000 mg/day and HCQ 400 mg daily [116]. In the third case, patients (*n =* 15) received MTX 0.2 mg/kg/week with prednisone 10 mg/day for 6 months [135]. In the fourth case, monotherapy or combination therapy with methotrexate, sulfasalazine and HCQ was used for 6 months (*n =* 40). Additionally, some patients received an anti-TNF agent (*n =* 16) [117]. In the latter case, adiponectin levels were significantly elevated in the third month of therapy with MTX 10–15 mg/week and prednisolone 7.5–15.0 mg/day (*n =* 65) [134]. Only one study (*n =* 46 early RA) reported a slight decrease in adiponectin levels after 6 months of DMARD treatment (MTX median dose 17.5 mg/week, sulfasalazine and leflunomide at standard doses) [36]. Additionally, one report reported no changes in circulating adiponectin levels after treatment with one or more of the following DMARDs: azathioprine, HCQ, leflunomide, MTX, SSA with prednisone at doses of 10 mg/day or less after 6 months (*n =* 127), 1 year (*n =* 91) and 2 years [137]. The great variety of different drugs in this group and their use in combination therapy make it difficult to investigate associations with clinical manifestations.

### 3.4. Tocilizumab

Tocilizumab (TCZ), a monoclonal antibody against the interleukin-6 receptor, in monotherapy or combination is recommended for treating moderate to severe RA, which had insufficient response to other DMARDs or to anti-TNF treatment [138].

In RA patients treated with TCZ, most studies report that adiponectin levels increased significantly after treatment as shown in Table 4. The only decrease was shown in one study where adiponectin was measured at 4 months, while in the other studies a time point of 6 months was used.

Studies also suggest TCZ may have protective role on CVD in RA patients, as beneficial changes in lipid profile are observed after treatment in most cases, but no correlation was found with adiponectin [27,121,140].

### 3.5. JAK Inhibitors

Janus kinases (JAKs) are enzymes transducing pro-inflammatory cytokine signals in cells, which contribute to an immune or inflammatory response in the cell. Their inhibition therefore leads to anti-inflammatory effects [142]. Baricitinib, a JAK 1/2 inhibitor, has been recommended for RA patients who do not respond to initial treatment with MTX or other conventional synthetic DMARDs [132]. In a recent study published on RA, adiponectin levels decreased significantly in patients treated with baricitinib monotherapy at 4 mg/day (*n =* 11) or in combination with other DMARDs (*n =* 4) for 6 months. The decrease in systemic inflammation was observed but not studied for adiponectin association [143].

### 3.6. Interleukin (IL)-1-Receptor Antagonists

Another important pro-inflammatory cytokine, IL-1, is targeted with biologic therapy. Patients treated daily with 100 mg of anakinra (*n =* 15), an interleukin-1 receptor antagonist, showed a significant improvement in type 2 diabetes-related metabolic parameters, but no changes in serum adiponectin concentrations over a treatment period of 6 months [124].

### 3.7. Anti-Interleukin-17A

IL-17 is an important pro-inflammatory cytokine produced by T helper 17 (Th17) cells upon stimulation with IL-23. In PsA patients, an increased number of polyfunctional circulating Th17 memory cells was found that produced IL-17 [144]. Adiponectin was shown to decrease the synthesis of IL-17 by acting on murine γδ-T-cells and human CD4+ and CD8+ T-cells, which lead to decreased inflammation in the skin [145]. Furthermore, lymphocytes from myelin-immunized adiponectin-deficient mice produced higher amounts of IL-17, which decreased after treatment with globular adiponectin [146]. The entanglement of adiponectin and the IL-17 signaling pathway led to the question of how inhibition of IL-17 could result in circulating adiponectin levels. Patients with PsA (*n =* 28) treated with secukinumab, a monoclonal antibody that binds to the protein IL-17A, with doses of 75 to 150 mg per month, showed no effect on serum adiponectin levels after the first, third and sixth month of therapy [147].

### 3.8. Cyclophosphamide

Cyclophosphamide (CY) metabolic intermediates are alkylating agents that cross-link DNA, causing cell death, the modulation of lymphocytes and the impairment of inflammatory responses [148]. CY’s main active metabolite, phosphoramide mustard, is produced in cells with low aldehyde dehydrogenases (ALDH) enzyme, thus enabling relatively cell-specific effects [149]. The expression of ALDH1A1 is abundant in adipose tissue and lungs, while expression is low in cultured fibroblasts, whole blood, heart, and skin [150], making them susceptible to CY effects [151], but leaving adipocytes, the main producers of adiponectin, unaffected.

CY is the most common treatment for SSc-ILD [152], based on a study that showed a significant improvement in forced vital capacity (FVC) in patients receiving cyclophosphamide compared to a placebo as a percentage of predicted FVC after a one-year follow-up [153]. Serum adiponectin levels in a group of SSc patients with active ILD (*n =* 8) increased more than five-fold after completion of three to six courses of intravenous pulse cyclophosphamide therapy and correlated significantly with a decrease in ILD [154].

### 3.9. n-3 Fatty Acids

Among other beneficial effects on cardiovascular health, n-3 fatty acids increase adiponectin levels by activating peroxisome-proliferator-activated receptor gamma (PPAR-γ) [155]. CVD is one of the major causes of death in SLE [156]. Two studies have been published on the effects of dietary supplements containing n-3 fatty acids on adiponectin levels in SLE, both using the same dose, 1800 mg eicosapentaenoic acid and 1200 mg docosahexaenoic acid daily. The 12-week study (SLE *n =* 22, HC *n =* 27) found no changes in adiponectin levels [157], while the 17-week study (SLE *n =* 41, HC *n =* 21) reported a significant increase and improvement in the disease activity score but did not include an adiponectin correlation [158].

To summarize, while GC, TCZ, CY and DMARD therapies are generally associated with the upregulation of adiponectin levels, anti-TNF therapy does not appear to cause changes in serum adiponectin levels, although these drugs successfully down-regulate inflammation. This may be partially due to the crosstalk and similarities in the structure of adiponectin and TNF-α. However, no binding of anti-TNF to adiponectin has been observed, so possible underlying mechanisms that could be related to signalization pathways will need to be investigated.

## 4. Divergent Changes in the Activation or Suppression of Adiponectin Gene Regulation by Transcription Factors Can Affect Its Serum Levels

Adiponectin promoters consist of nuclear receptor sites (PPARG, LRH, RXR), transcription factor sites that enhance adiponectin gene expression (C/EBPα, SREBP1c, TFAP2B, FOXO1, SP1) and co-regulators of transcription factors (SIRT1, NCOR1, NCOR2), among others (Figure 6) [159].

### 4.1. PPAR-γ

Adiponectin expression is best characterized by PPAR-γ regulator, a ligand activated by binding to unsaturated fatty acids and eicosanoids such as 15-deoxy-Δ12,14-prostaglandin J2, TZD, and cannabinoids. PPAR-γ heteromerizes with the retionid X-receptor and activates adiponectin expression. Contrary to expectations, in SARDs with high adiponectin serum levels, such as RA, low levels of PPAR-γ expression or activity is present and also beneficial effects of PPAR-γ activation have been demonstrated [160,161]. This indicates that adiponectin serum levels are elevated in these disorders for reasons other than PPAR-γ activation. In SLE, elevated PPAR-γ levels have been shown [162], but the beneficial effects of PPAR-γ activation have also been demonstrated in cell experiments and animal models [163,164]. The PPAR-γ activators TZDs (pioglitazone and rosiglitazone) are approved by the FDA for type 2 diabetes [165], but are thought to exert beneficial effects on SARDs with altered PPAR-γ activity [160]. Clinical trials with TZDs have been conducted in RA and in SLE patients, with the focus on inflammation control, cardiovascular protection and skeletal muscle dysfunction [165]. In vitro and in vivo studies have shown positive effects of PPAR-γ activators also in SSc skin and lung fibrosis, as well as in many other fibrotic disorders [166]. However, the clinical study with IVA337—Lanifibranor, a pan-PPAR agonist, has failed to demonstrate a significant improvement in SSc treatment [165]. This means that other factors (PPAR and non-PPAR related) that contribute to the pathology of these diseases also play an important role in regulating adiponectin levels. One of these factors could be the PPAR-γ nuclear corepressor NCoR, which is aberrantly activated in SSc skin [167].

### 4.2. Id3

Id3 is a repressor of adiponectin expression that acts on SREBP, an adiponectin transcription promoter in adipocytes [15,168]. It has been found upregulated in lung tissue and fibroblasts of SSc and idiopathic pulmonary fibrosis, thereby maintaining fibroblasts in a dedifferentiated, hyperproliferative and apoptosis-resistant state [169]. Id3-deficient mice develop an autoimmune disease similar to human SS [170], but Id3 has been shown to be elevated in RA synovium [171] and correlates with the SLEDAI in SLE [172].

### 4.3. ATF3

Another repressor of adiponectin gene expression, activating transcription factor 3 (ATF3), suppresses the expression of inflammatory cytokines/chemokines in immune cells after various stimuli [173]. ATF3 was elevated in skin/fibroblasts of SSc, in association with the profibrotic signal transduction of TGF-β [166].

### 4.4. SIRT1, FoXO1 and C/EBPα

Experiments suggest that C/EBPα is required to fully activate adiponectin gene expression, although the physiological significance is not clear. FoXO1 binds to C/EBPα and its activity is regulated by insulin, IGF-1 and SIRT1. Adiponectin gene expression induced by SIRT1 is likely to occur through FoxO1 deacetylation [15]. Overexpression of SIRT1 is found in synovial RA tissue [174] and in LN [175], but SIRT1 has been shown to be reduced in peripheral blood mononuclear cells of SSc patients with pulmonary fibrosis and in lung tissues of bleomycin-induced lung fibrosis mice [176]. This could be an additional explanation for diverging adiponectin levels in SARDs.

## 5. How Inflammatory Cytokines IL-6 and TNF-α Regulate Adiponectin Levels in SARDs

Inflammatory cytokines, IL-6 and TNF-α, downregulate adiponectin levels [177,178,179], which was demonstrated at the mRNA and protein level in 3T3-L1 adipocytes and in a mouse model. Adiponectin gene expression was dose- (0.5–100 ng/mL) and time-dependent (0–24 h) on IL-6 stimulation [177,178,179]. Paradoxically, in some SARDs, such as RA and SLE, adiponectin levels remain elevated [4,6,88] despite persistent inflammation and higher concentrations of IL-6 and TNF-α [4,6,88,138,180,181]. Of note, the IL-6–TNF-α-adiponectin relation has been also observed in the opposite direction: adiponectin reduces the release of IL-6 and TNF-α in adipocytes and stromal-vascular cells [11].

Both TNF-α and IL-6 inhibit adiponectin synthesis by downregulating PPAR-γ activity. TNF-α activates JNK that inhibits PPAR-γ DNA binding trough phosphorylation of PPAR-γ, and shares a pathway with IL-6 through the STAT-SIRT1-FoxO1 pathway. This inhibits occupancy of peroxisome proliferator response element (PPRE) trough interaction of FoxO1 and PPAR-γ [182,183]. IL-6, on the other hand, also inhibits adiponectin expression through the Erk1/2—NFAT pathway, which is PPAR-γ unrelated (Figure 6) [15]. Differences in PPAR-γ activation and the non-PPAR-γ dependent regulation of adiponectin expression could explain why we observe a serum adiponectin increase after anti-IL-6 treatment, but no change after an anti-TNF treatment in SARDs.

Adiponectin in certain SARDs is produced abundantly in cells other than adipocytes, such as synoviocytes in RA [59]. Some of these cell types have insignificant PPAR-γ expression, which could be the reason that IL-6 and TNF-α, through suppression of PPAR-γ activity, cannot sufficiently decrease serum adiponectin levels. PPAR-γ is the product of one gene, but has been reported to have 2 splice variants. While PPAR-γ1 is present in macrophages, colonic epithelium, endothelium and vascular smooth muscle cells, PPAR-γ2 is predominantly present in adipose tissue [160]. PPAR-γ expression is increased in adipogenesis and in monocytes, when transdifferentiating into macrophages [184]. Interestingly, RNA expression is very low in normal, unstimulated circulating blood cells [151], although the PPAR-γ gene was originally cloned from bone marrow cDNA library [185]. Activation of PPAR-γ promotes both M2 polarization in macrophages and Th2 polarization in T cells [186] and in RA, which lacks PPAR-γ activity. M1 and Th1 are the predominant cell types.

In summary, first, the lack of PPAR-γ activity in RA and SLE was proven in vitro and in vivo models, minimizing the ability of TNF-α and IL-6 to downregulate adiponectin levels via this pathway. Second, TNF-α- and IL-6-related adiponectin downregulation was mainly characterized in cells expressing high levels of PPAR-γ, and could be efficient in obesity, for example. However, this mechanism is not as efficient in RA or SLE, where serum adiponectin levels depend on synthesis in other cells with intrinsically low PPAR-γ and thus cannot be further downregulated via this pathway. Altogether, this points to the reason that TNF-α and IL-6 do not inhibit serum adiponectin levels in RA and SLE patients.

## 6. Conclusions

The deregulation of adiponectin in SARDs was seen in both upregulated and downregulated mechanisms. We concluded that diseases with less prominent inflammation (SSc), where adipocytes have been shown to be reduced at the site of disease pathology, have decreased adiponectin levels and a negative correlation with disease activity. On the other hand, highly inflammatory diseases (RA, SLE) show increased adiponectin levels and a positive correlation with clinical manifestations. Treatment used in SARDs is associated with either decreased adiponectin levels, as with JAK inhibitors, or increased levels, as in cases where GC, CY, TCZ and DMARDs are used (Figure 7). The paradox of why increased adiponectin levels are present in RA and SLE despite high levels of proinflammatory cytokines IL-6 and TNF-α could be explained at the level of adiponectin gene regulation. In general, IL-6 and TNF-α inhibit adiponectin via the suppression of PPAR-γ, which is highly expressed in adipocytes. However, in some SARDs, adiponectin is also abundantly produced by other cells that lack PPAR-γ activity, leaving adiponectin production intact. Differences in PPAR-γ activation, as well as other non-PPAR-γ-dependent signaling pathways regulating adiponectin expression, may provide an answer to why adiponectin levels are associated with anti-IL-6 treatment, but not with anti-TNF therapy. However, the molecular mechanisms need further investigation.

In SARDs, where elevated adiponectin levels are associated with deleterious manifestations, blocking adiponectin with antagonists (ADP400) [179] and adiponectin-targeting agents (monoclonal antibodies KH7-33, KH4-8) is suggested. In contrast, various AdipoR agonists (AdipoRon, ADP355) and PPAR-γ activators (thiazolinediones) could be useful in SARDs that lack adiponectin. In general, it seems very important to ensure normal or optimal levels of adiponectin and its activity, as any deregulation is associated with many pathologies.

## Figures and Tables

**Figure 1 ijms-22-04095-f001:**
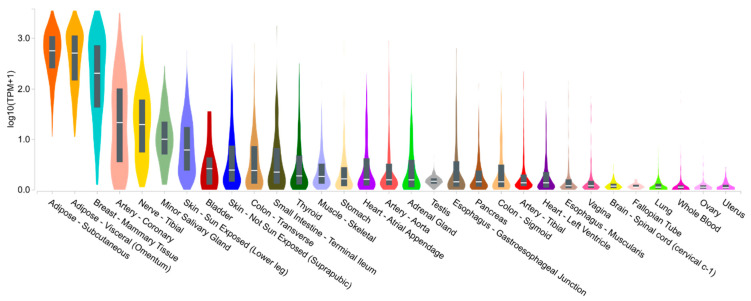
Gene expression of adiponectin—ADIPOQ (ENSG00000181092.9) in human tissues from the Genotype-Tissue Expression (GTEx) portal. The expression values are given in transcripts per million (log10 (TPM + 1)), calculated from a gene model with isoforms collapsed to a single gene. No other normalization steps were applied. Box plots are shown as median and as 25th and 75th percentile. Only tissues with the median TPM > 0.1 are presented.

**Figure 2 ijms-22-04095-f002:**
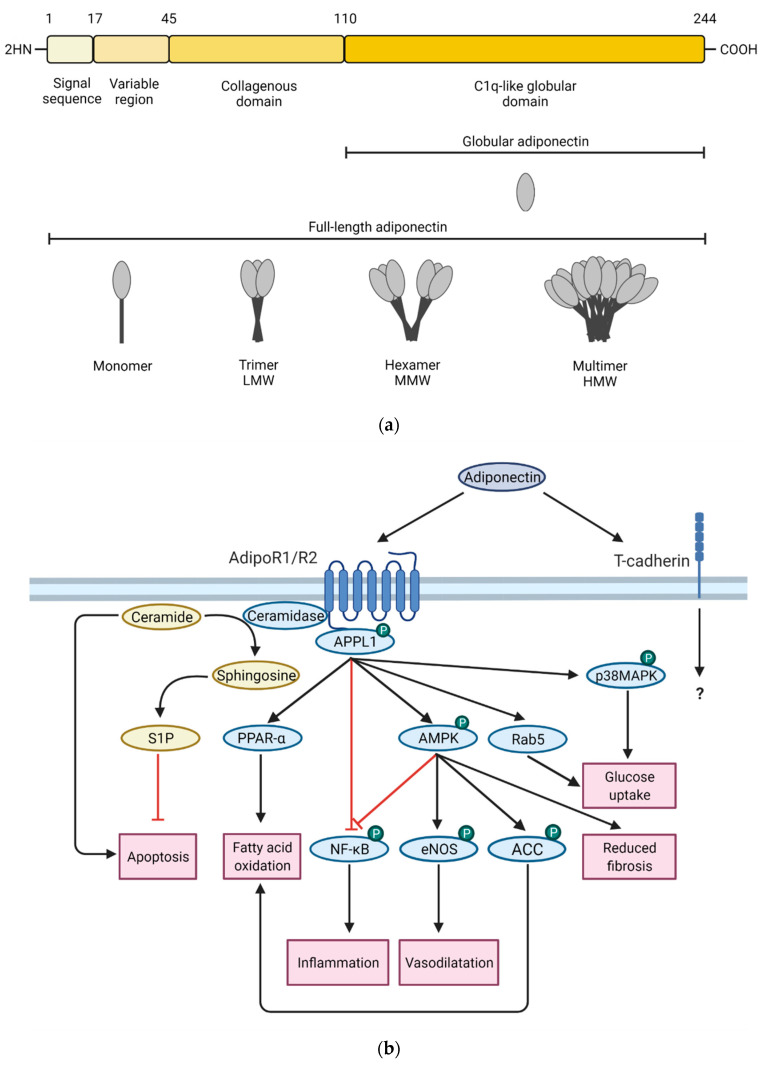
(**a**) Adiponectin structure and isoforms. (**b**) Simplified adiponectin downstream signaling. ACC: Acetyl-CoA carboxylase, AMPK: AMP-activated protein kinase, APPL1: adaptor protein, phosphotyrosine interacting with PH domain and leucine zipper 1, eNOS: Endothelial NOS, NF-Κb: Nuclear factor-κB, p38MAPK: p38 mitogen-activated protein kinase, PPAR-α: Peroxisome proliferator-activated receptor α, S1P: sphingosine-1-phosphate.

**Figure 3 ijms-22-04095-f003:**
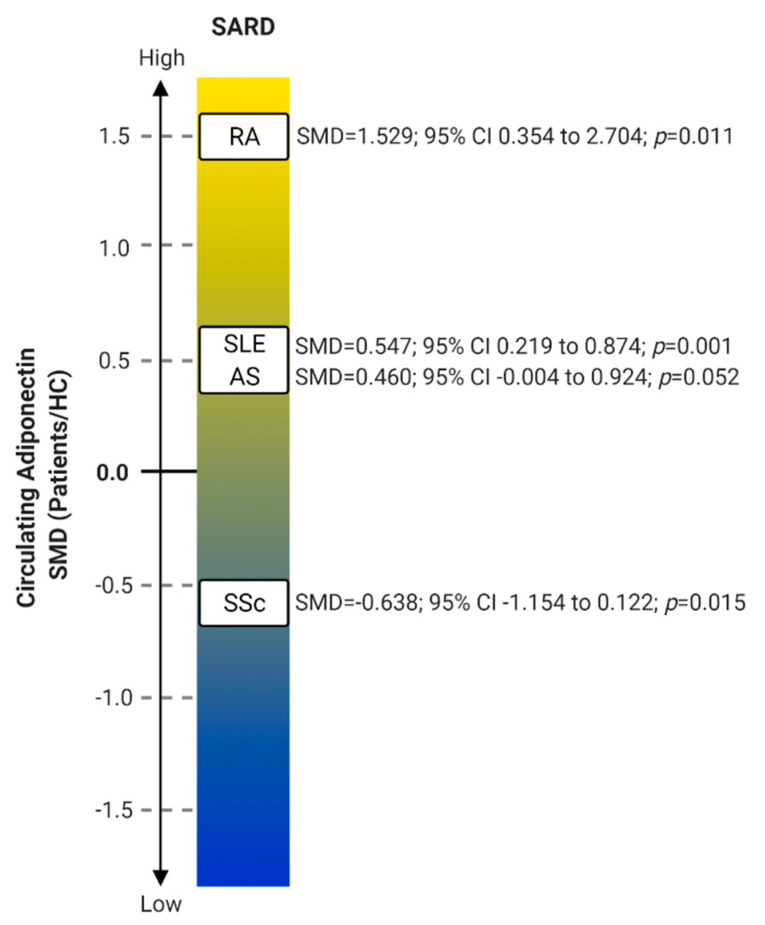
Adiponectin circulating levels observed in certain SARDs. The values are presented as standard mean difference (SMD) with 95% confidence intervals (CI) and association *p* values. The most recent meta-analysis with the highest number of studies included was presented for each SARD. AS: Ankylosing spondylitis, HC: Healthy controls, RA: Rheumatoid arthritis, SLE: Systemic lupus erythematosus, SSc: Systemic sclerosis.

**Figure 4 ijms-22-04095-f004:**
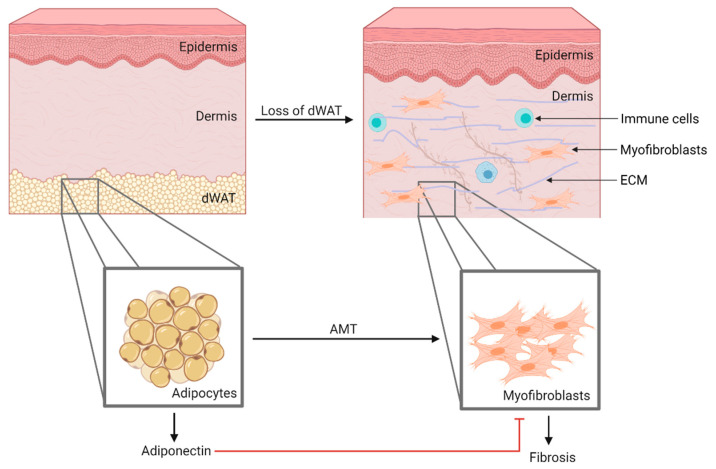
Local loss of dWAT in SSc skin in addition to adipocytes undergoing AMT causes decreased adiponectin secretion. Consequently, its inhibitory effects on myofibroblasts are lost, which results in fibrosis. AMT: adipocyte mesenchymal transition, dWAT: dermal white adipose tissue, EMC: Extracellular matrix, SSc: Systemic sclerosis.

**Figure 5 ijms-22-04095-f005:**
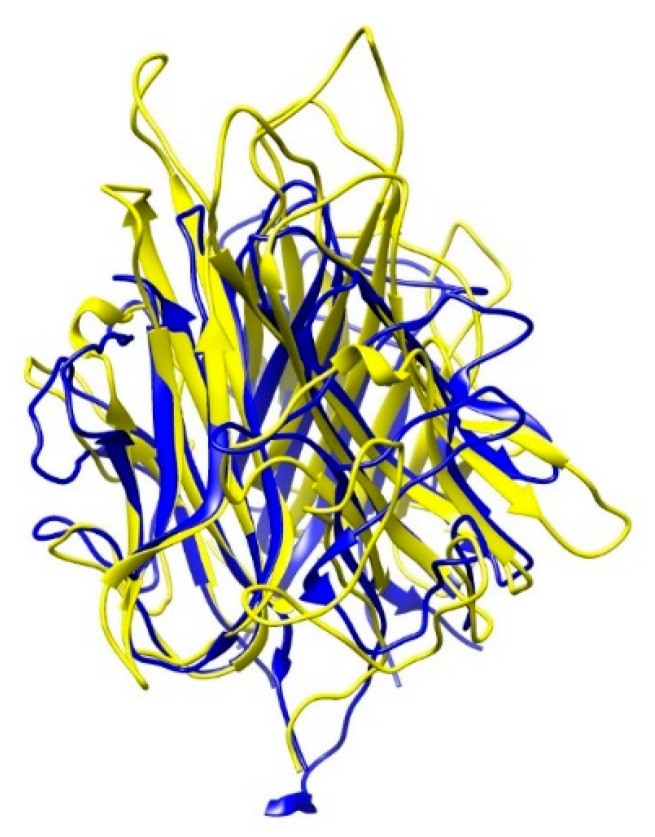
Superposition of trimeric globular domain of adiponectin (PDB: 6U66) in blue and TNF-α (PDB:1TNF) in yellow shows high 3D-structure similarity.

**Figure 6 ijms-22-04095-f006:**
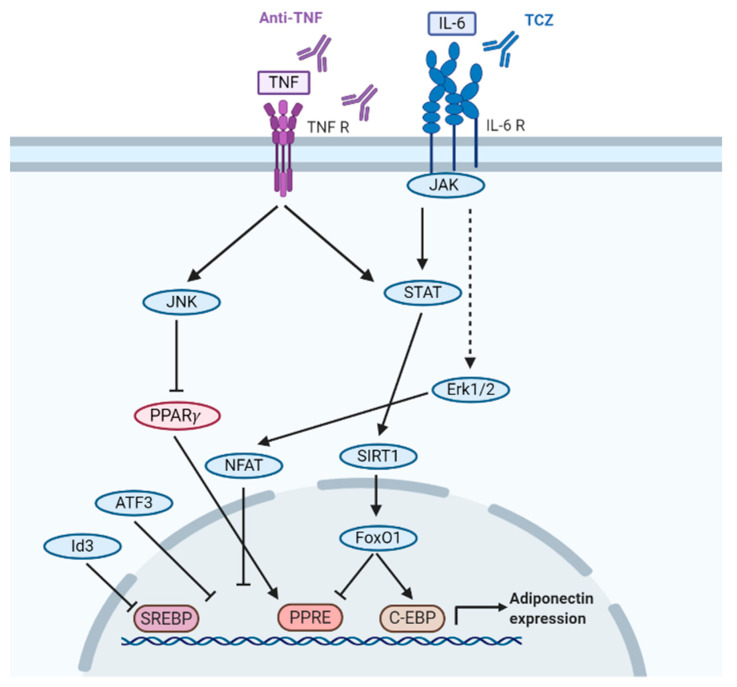
Simplified ADIPOQ gene regulation with TNF-α and IL-6. ATF3: Activating Transcription Factor 3, C-EBP: CCAAT-enhancer-binding protein, Erk1/2: extracellular signal-regulated kinases 1/2 FoxO1: Forkhead box protein O1, Id3: Inhibitor Of DNA Binding 3, IL: Interleukin, JAK: Janus kinase, JNK: Jun N-terminal kinase, NFAT: Nuclear factor of activated T-cells, PPAR-γ: Peroxisome-proliferator-activated receptor gamma, PPRE: Peroxisome proliferator response element, R: receptor, SIRT: Sirtuin, SREBP: Sterol regulatory element-binding protein, STAT: signal transducer and activator of transcription, TNF: Tumor necrosis factor.

**Figure 7 ijms-22-04095-f007:**
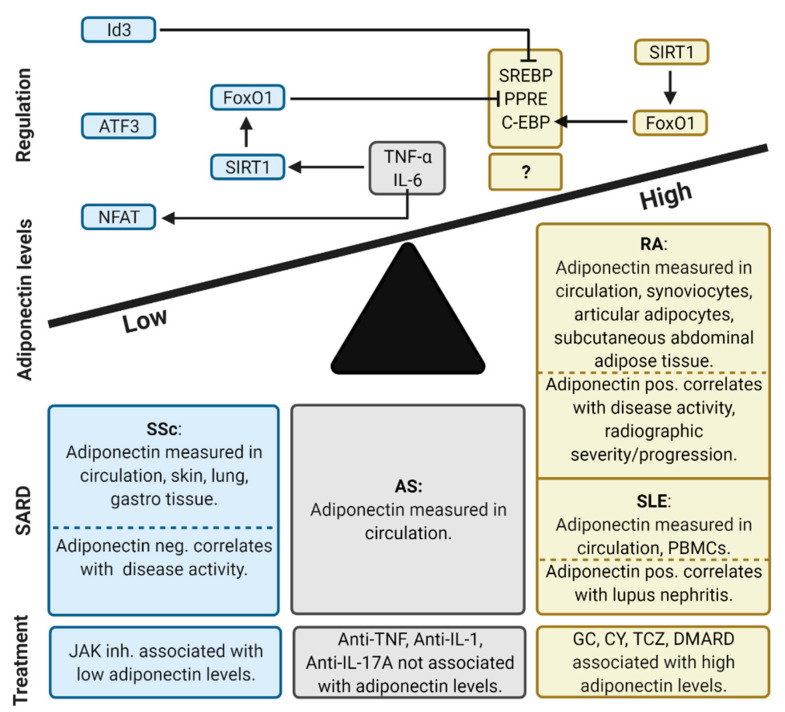
Summary of adiponectin levels in SARDs associated with clinical manifestations, treatment and adiponectin gene regulation. AS: Ankylosing spondylitis, ATF3: Activating Transcription Factor 3, C-EBP: CCAAT-enhancer binding protein, CY: Cyclophosphamide, DMARD: Disease-modifying antirheumatic drugs, FoxO1: Forkhead box protein O1, GC: Glucocorticoids, Id3: Inhibitor Of DNA Binding 3, IL: Interleukin, JAK: Janus kinase, NFAT: Nuclear factor of activated T-cells, PBMCs: Peripheral blood mononuclear cells, PPRE: Peroxisome proliferator response element, RA: rheumatoid arthritis, SIRT: Sirtuin 1/NAD-dependent deacetylase, SLE: Systemic lupus, erythematosus, SREBP: Sterol regulatory element-binding protein, SSc: Systemic sclerosis, TCZ: Tocilizumab, TNF: Tumor necrosis factor.

**Table 1 ijms-22-04095-t001:** Clinical observations associated with circulating adiponectin levels in RA patients.

Clinical Feature	RA Patients(N)	Serum Adiponectin Levels Associations/Correlations	Reference
Disease activity	351/90/52	Positive association with DAS28-ESR.	[26,38,42]
	51	Early RA with high adiponectin levels was less likely to have MHAQ score > 3 and RAPID3 score > 12.	[29]
	121	Stratifying according to DAS28 (low, moderate and high activity), there were no differences seen for adiponectin.	[43]
	180	Negative correlation of total, HMW, MMW, and LMW adiponectin with the DAS28.	[44]
	40	Negative correlation with DAS28.	[45]
	70	Positive correlation with DAS28-ESR in active disease.	[46]
	80	Negative correlation with the number of swollen joints.	[47]
Radiographic severity/progression	324	Positive association of total, but not HMW adiponectin with radiographic progression.	[48]
	242	Positive correlation with radiographic severity.	[49]
	632	Independent association with baseline total SHS, ΔSHS ≥ 1 and predicted ΔSHS ≥ 5.	[36]
	253	Positive association with radiographic progression over 4 years.	[50]
	152	Positive association with radiographic progression.	[51]
CV-related	54	No correlation with coronary artery calcification.	[52]
	210	In RA patients with abdominal obesity or no clinically evident joint damage associated with decreased carotid atherosclerosis.	[53]
	192	Leptin: adiponectin ratio associated with common carotid artery resistive index.	[54]
	210	Positive associations of total and HMW adiponectin with increased blood pressure parameters, and in white patients additionally with endothelial activation.	[55]
Bone-related	112	Negative association with trabecular volumetric bone mineral density and cortical thickness.	[56]
	38	Positive correlation with osteopontin in serum.	[28]
Muscle-related	50	Negative association with appendicular lean mass index and muscle cross-sectional area.	[57]

DAS28: Disease Activity Score of 28 joints, ESR: erythrocyte sedimentation rate, HMW: High-molecular-weight, LMW: Low-molecular-weight, MHAQ: Multidimensional Health Assessment Questionnaire, MMW: Medium-molecular-weight, RA: Rheumatoid arthritis, RAPID3: Routine Assessment of Patient Index Data 3, SHS: Sharp-van der Heijde Score.

**Table 2 ijms-22-04095-t002:** Anti-TNF effects on circulating adiponectin levels in rheumatoid arthritis, psoriatic arthritis and ankylosing spondylitis.

SARD	Patients (N)	Anti-TNF	Treatment Regimen	Study Duration	Influence on Adiponectin Levels	Ref.
RA	16	adalimumab	40 mg every 2 weeks	1 year	No change	[118]
		etanercept	25 mg twice a week			
		Infliximab	3 mg/kg every 8 weeks			
	48	adalimumab	40 mg every 2 weeks	16 weeks	No change	[119]
	171	adalimumab	40 mg every 2 weeks	16 weeks	No change	[120]
	96	adalimumab	at approved doses	24 weeks	No change	[121]
		certolizumab	at approved doses			
		etanercept	at approved doses			
		infliximab	at approved doses			
	8	adalimumab	40 mg every 2 weeks	2 years	No change	[122]
		etanercept	50 mg every week			
		infliximab	3 mg/kg			
	21	adalimumab	at approved doses	12 weeks	No change	[123]
		certolizumab	at approved doses			
		etanercept	at approved doses			
		golimumab	at approved doses			
		infliximab	at approved doses			
	15	anti-TNF	at approved doses	6 months	No change	[124]
	33	adalimumab	40 mg every 2 weeks	12 and	Increase	[115]
		etanercept	50 mg every week	24 weeks		
		infliximab	5 mg/kg every 8 weeks			
	16	infliximab	3 mg/kg in weeks 0, 2 and 6 and every 8 weeks after	24 months	No change	[116]
				3–12 months	Increase	
	16	adalimumab	-	6 months	Increase	[117]
		eternacept	-			
		infliximab	-			
PsA	126	onercept	50 mg or 100 mg three times a week	12 weeks	No change	[120]

	405	golimumab	50 mg or 100 mg every 4 weeks	14 weeks	No change	[125]

AS	30	infliximab	5 mg/kg in weeks 0, 2, 6 and every 8 weeks after	6 months	No change	[126]

	29	infliximab	Infusion (120 min)	before and right after	No change	[127]

	12	adalimumab	40 mg every 2 weeks	2 years	No change	[122]
		etanercept	50 mg every week			
		infliximab	5 mg/kg			

AS: Ankylosing spondylitis, PsA: Psoriatic arthritis, RA Rheumatoid arthritis.

**Table 3 ijms-22-04095-t003:** Glucocorticoids (prednisolone) effects on circulating adiponectin levels in rheumatoid arthritis.

Patients (N)	Treatment Regimen	Additional Therapy	Additional Therapy Regimen	Study Duration	Influence on APN Levels	Ref.
65	7.5–15.0 mg/day	MTX	10–15 mg/week	3 months	Increase	[134]
15	10 mg/day	MTX	0.2 mg/kg/week	6 months	Increase	[135]
15	10 mg/day	MTX + ATV	0.2 mg/kg/week 40 mg/day	6 months	Increase	[135]
9	60 mg/day (week 1); 40 mg/day (week 2)	-	-	2 weeks	Increase	[119]
19	Tapered high dose: 60 mg/day (week 1); 40 mg/day (week 2); 30 mg/day (week 3); 20 mg/day (week 4); 15 mg/day (week 5); 10 mg/day (week 6); 7.5 mg/day (thereafter)	HCQ, SSA MTX	400 mg/day 2 g/day 10 mg/week	22 weeks	No change	[119]
127	10 mg/day or less	DMARD	Stable therapy	6 months	No change	[137]
91				1 year		
52				2 years		

ATV: Atorvastatin, DMARD: Disease-modifying antirheumatic drugs, HCQ: Hydroxychloroquine, MTX: Methotrexate, SSA: Sulfasalazine.

**Table 4 ijms-22-04095-t004:** Tocilizumab effects on circulating adiponectin levels in rheumatoid arthritis.

N of Patients	Treatment Regimen	Additional Therapy	Study Duration	Influence on APN Levels	Ref.
41	8 mg/kg	± MTX, GC, NSAID	6 months	Increase	[141]
40	8 mg/kg every 4 weeks	± MTX, SSA, HCQ, Leflunomide, GC, statins, anti-diabetics	4 months	Decrease	[27]
20	8 mg/kg every 4 weeks	± NSAID, coxibs, GC	6 months	Increase	[140]
24	8 mg/kg every 4 weeks	MTX (± NSAID, coxibs, GC)	6 months	Increase	[140]
47	-	-	6 months	Increase	[121]

GC: Glucocorticoids, HCQ: Hydroxychloroquine, MTX: Methotrexate, NSAID: Nonsteroidal anti-inflammatory drugs, SSA: Sulphasalazine.

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
