# Peer review of "Adiponectin Deregulation in Systemic Autoimmune Rheumatic Diseases"

_ijms, 2021, doi:10.3390/ijms22084095_

Round 1

Reviewer 1 Report

In this review, the authors summarized the relationship of adiponectin in systemic autoimmune rheumatic diseases (SARD), and reviewed the effects of various SARD drugs on adiponectin levels. Finally, the authors presented some possible explanations why adiponectin is deregulated in the context of therapy and gene regulation. Considering the quality of this manuscript, I would like to recommend it for publication after some revisions. The detailed comments are as follows:

  1. Is this manuscript distinguishable from other recent reviews on this topic? Please explain the difference between this manuscript and other review articles in this field.
  2. I think the author listed a variety of diseases related to the expression of adiponectin in the second part of the manuscript, and I think the author should briefly describe the mechanism of adiponectin-induced diseases.

Reviewer 2 Report

The manuscript is a very elegant review on the role of adiponectin and thereby it´s implications for the adipogenesis- infammatory axis in rheumatoid arthritis and other systemischer diseases. The author present a will written and perfectly illustrated overview. There are no comments.

Author Response

We thank the Reviewer 2 for his positive feedback on our manuscript.

Round 2

Reviewer 1 Report

It can be accented  now.